# The Effects of Competition on Exercise Intensity and the User Experience of Exercise during Virtual Reality Bicycling for Young Adults [note 1]

**DOI:** 10.3390/s24216873

**Published:** 2024-10-26

**Authors:** John L. Palmieri, Judith E. Deutsch

**Affiliations:** 1RiVERS Lab in the Department of Rehabilitation & Movement Sciences, Rutgers School of Health Professions, Newark, NJ 07107, USA; deutsch@shp.rutgers.edu; 2Rutgers School of Graduate Studies, Newark, NJ 07103, USA; 3Rutgers New Jersey Medical School, Newark, NJ 07103, USA

**Keywords:** virtual reality, bicycling, aerobic exercise, wearable sensors, visual feedback, competition, eye-tracking, visual attention, motivation, enjoyment

## Abstract

Background: Regular moderate–vigorous intensity exercise is recommended for adults as it can improve longevity and reduce health risks associated with a sedentary lifestyle. However, there are barriers to achieving intense exercise that may be addressed using virtual reality (VR) as a tool to promote exercise intensity and adherence, particularly through visual feedback and competition. The purpose of this work is to compare visual feedback and competition within fully immersive VR to enhance exercise intensity and user experience of exercise for young adults; and to describe and compare visual attention during each of the conditions. Methods: Young adults (21–34 years old) bicycled in three 5 min VR conditions (visual feedback, self-competition, and competition against others). Exercise intensity (cycling cadence and % of maximum heart rate) and visual attention (derived from a wearable eye tracking sensor) were measured continuously. User experience was measured by an intrinsic motivation questionnaire, perceived effort, and participant preference. A repeated-measures ANOVA with paired *t*-test post hoc tests was conducted to detect differences between conditions. Results: Participants exercised at a higher intensity and had higher intrinsic motivation in the two competitive conditions compared to visual feedback. Further, participants preferred the competitive conditions and only reached a vigorous exercise intensity during self-competition. Visual exploration was higher in visual feedback compared to self-competition. Conclusions: For young adults bicycling in VR, competition promoted higher exercise intensity and motivation compared to visual feedback.

## 1. Introduction

The guidelines for physical activity and exercise in the United States recommend 150 min of moderate–high intensity aerobic exercise per week for adults [1]. Adhering to this guideline can help people maintain a healthy weight, lower high blood pressure, and delay the onset of several chronic diseases [1,2,3]. Despite the known benefits of exercise, 20–25% of Americans aged 18–39 do not meet this requirement, with adherence among young adults decreasing with age [4]. Commonly reported barriers to explain these low adherence rates are reports of exercise being boring and inconvenient and physical factors such as discomfort during and after exercise [5]. A group of undergraduate students in Canada (average age 23 years old) reported time commitments, lack of available resources, lack of interest, and cost among their top barriers [6]. Considering these barriers, factors like motivation and enjoyment may be important to consider in promoting adherence to high-intensity exercise.

Enhancing motivation to exercise may help reduce the known barriers to exercise, improve adherence to exercise, and help individuals achieve the established benefits of regular exercise. Intrinsic motivation has been defined in Ryan and Deci 2000 as “doing an activity for the inherent satisfaction of the activity itself” [7]. As intrinsic motivation is directly related to the task being performed, the task (exercise) can be modified to foster intrinsic motivation. Self-determination theory (SDT) underlies intrinsically motivated behavior and highlights different elements or properties of a task that could foster intrinsic motivation [8,9]. The specific reasons behind one’s motivation to perform an action fall along a spectrum of self-determination, whereby high levels of self-determination are associated with the perception of having a choice or agency in deciding to perform the action. The level of self-determination in the motivation behind a behavior is associated with how robust the behavior is (adherence). Self-determination is closely tied to intrinsic motivation, whereby an action is pursued because the action itself is inherently enjoyable regardless of other associated outcomes or external consequences of the action. Intrinsically motivated behaviors are those with the highest degrees of self-determination, as an individual performing the task finds it inherently interesting or enjoyable [9,10].

Virtual reality (VR) has been proposed as a tool to facilitate exercise and improve motivation. VR uses visual and auditory stimuli as well as feedback promoting interaction with the computer-generated simulation, that may translate to enhanced enjoyment and motivation to exercise. These interactive elements provide information to users that is relevant to the task and may motivate them to exercise at higher intensities for longer periods of time [11,12,13,14].

In addition to visual feedback, competition in VR may provide an additive effect for promoting intense exercise and increasing motivation. Studies in healthy adults have shown that using competitive stimuli in VR can enhance exercise intensity, motivation, and enjoyment [15,16]. Specifically, during a VR bicycling task where participants were instructed to compete against their own performance on previous trials, exercise intensity and motivation were higher compared to a non-competitive condition [17]. Healthy young adults in an 8-session (over four weeks) study competed against bicycle riders (virtual opponents) that represented their previous best times. In each session, participants were instructed to perform better than the trial from their previous session. Ghost avatar cyclists pedaled along the track representing participants’ times from previous sessions, a form of competition based on reference to one’s own performance and self-modeling feedback [11]. In the control group, participants bicycled in the same virtual environment (VE) but without other riders on the road. Participants exercised at a higher intensity (faster cadence) and enjoyed the activity more compared to a control group that did not compete against their previous best trial. Self-competition improved task-related performance and motivation for healthy young adults as assessed across several domains of IMI. Therefore, self-competition may promote intrinsically motivated behavior and enhance exercise performance [17].

This motivation to exercise at a higher intensity in competitive settings may depend on several personal characteristics, including competitiveness, a trait that is often associated with outperforming another person in a wide range of competitive scenarios, ranging from sports to tasks in academic or work settings. Competitive profiles have been explored, identifying different types of competitiveness, such as self-developmental, anxiety-driven, and hyper-competitiveness, which may be important to consider [18]. These competitive profiles are related to individual perceptions of competition and may provide valuable insight into understanding how motivated an individual will be to engage in competitive exercise. However, there are only a handful of studies reporting on motivation to exercise in the context of competitive profiles, with a dearth of research specifically investigating the specific profiles described above [18].

In addition to intrinsic motivation, visual attention may serve to explain behavior during exercise in VR, particularly as an underlying mechanism that may drive exercise intensity [19]. Eye movements have been closely linked to spatial attention, primarily in theories of selective attention, whereby individuals focus on a particular area of a vast visual field, therefore selecting where to direct their attention. This continuous process of sampling a visual field based on attention to different elements of the field highlights the close link between visual attention and eye movements. The literature in this area supports the theory that the voluntary movement of the eyes requires a movement or shift in one’s spatial attention first. Wearable eye-tracking sensors embedded within VR headsets can be conveniently utilized to measure selective attention in pre-specified areas of the visual field [20,21]. These measures of visual attention may capture an external focus of attention during tasks. Though visual attention measured with eye-tracking sensors may probe the mechanism associating attention with exercise intensity during virtual cycling, it has not been reported in the literature. Understanding attention during VR exercise tasks may provide insight into how participants interact with the simulation and whether the stimuli are associated with exercise intensity [11].

The purpose of this study is to use a custom VR system implementing various sensors to gain insight into exercise intensity, motivation, enjoyment, and visual attention. Two different forms of competition will be studied, competition against others and competition against oneself (self-competition), to gain insight into how the perception of competition impacts exercise intensity, enjoyment, and motivation. We hypothesize that both forms of competition will produce higher neuromuscular and cardiovascular intensity, motivation, and enjoyment compared to visual feedback based on previous studies for healthy adults in the literature. According to SDT and the impact of intrinsic motivators, we also hypothesize that self-competition will produce higher exercise intensity, motivation, and enjoyment than competition against others. Through an exploratory aim, we hypothesize that attention on the task will be higher during competitive VR bicycling compared to visual feedback as a higher exercise intensity and more positive user experience of exercise may be related to a higher task-related focus of attention [22].

## 2. Materials and Methods

### 2.1. Participants and Sample Size Justification

An a priori power analysis was conducted using cycling power output data (Watts) published in Michael et al., 2020, comparing competitive to non-competitive bicycling for healthy adults in a fully immersive virtual reality [17]. The power analysis, conducted in G*Power (Version#3.1.9.5. Aichach, Germany) [23], determined that 17 participants would be required to reach a power level of 0.80 (Cohen’s f = 0.35; moderate effect size; correlation between repeated measures = 0.5). Accounting for a 30% attrition rate, 25 participants were recruited [24]. The study was approved by the Institutional Review Board at the Rutgers School of Health Professions (Pro2021002091) and was also registered at https://clinicaltrials.gov/study/NCT05253703 (accessed on 25 October 2024, NCT05253703). Participants were primarily recruited through ClinicalTrials.Gov and through IRB-approved flyers posted on bulletin boards throughout the Rutgers Biomedical Health Sciences campus in Newark, NJ, USA, which includes Rutgers New Jersey Medical School and the Rutgers School of Health Professions.

### 2.2. Participant Screening

All participants were screened for eligibility. The screening included readiness for activity using the Physical Activity Readiness Questionnaire (PARQ+ 2021) [25] and depression using the Patient Health Questionnaire (PHQ-9) [26]. Inclusion criteria were as follows: able to provide informed consent to participate in the trial, participant age between 21 and 44 years, and able to ride an exercise bicycle (stationary upright). Exclusion criteria were as follows: a history of severe pulmonary or cardiovascular disease, diabetes with elevated blood glucose (uncontrolled), any neurological disorder including a traumatic brain injury, not able to follow instructions or directions, blindness or deafness rendering individual unable to see or hear the VR simulation, any medical condition that is unstable, severe arthritis or other rheumatological or musculoskeletal disorders including those requiring recent total knee or hip surgery, any medical condition or illness that would render riding an exercise bicycle difficult for the individual, and depression of at least moderate severity (score of 10 or more on PHQ-9) [26].

### 2.3. Experimental Setup

Custom virtual cycling simulations were developed using the Unity 3D game engine (Version #: 2019.4.33f1, San Francisco, CA, USA). Three-dimensional (3D) assets were created using 3DS Max, Maya, as well as being purchased from the Unity store. The HTC Vive Unity developer kit allowed for the customization of the VE to display on the head-mounted display (HMD) with the utilization of functions included in Unity 3D for smooth camera motion when the participants rotated their heads to scan the VE. C# was used to program the game logic, interactions, and feedback in the VE, user interface (UI) controls, and read the Wahoo RPM Cadence Pod (Atlanta, GA, USA) at 1 Hertz (Hz) attached to the bike via an Adaptive Network Topology (ANT+) protocol. Cycling cadence data from the sensors were integrated into the program to control the speed of the bicycle in the VE being projected in the HMD. A custom algorithm using data buffers and smoothing was programmed to allow for smooth motion within the VE, as false zero values transmitted by the cadence sensor would have otherwise caused abrupt pauses in the cycling speed and simulation flow. Importantly, the environment was customizable as the exercise intensity targets were adjusted to each person’s performance. To facilitate this goal, cycling cadence was transmitted in real time to the UI, which allowed the researcher administering the program to monitor each participant’s cadence and interpret a live feed of the baseline cycling cadence. Heart rate (HR) was recorded continuously via an optical wristband sensor validated for moderate–high intensity exercises (Polar OH1, collected at 1 Hz) (Kempele, Finland) [27]. The HMD also included integrated wearable eye-tracking technology that captured eye movements continuously throughout all 3 cycling bouts (50–60 Hz). By tracking the position of the participant’s gaze data, visual attention on different areas of the visual field and VE could be characterized. The design of the simulations accounted for important features to include in VE design such as field of view [28], spatial frequency [29,30], color contrast [31], texture and scale of objects [28,31], and previous experience by the research team in designing VEs [14].

For all conditions, participants bicycled on a stationary upright bike, the COSMED Ergoselect 100 (Rome, Italy), with resistance set between 30 and 50 Watts. The resistance was adjusted based on participant preference and was held constant across all 3 conditions. Participants also chose whether they wanted to exercise with (*n* = 10) or without (*n* = 15) a facemask for protection against the spread of respiratory viruses, but this was consistent across all 3 trials for the given participant. See Figure 1 for an image of the experimental setup and Figure 2 for a flow diagram of the simulation.

### 2.4. Research Procedures

#### 2.4.1. Baseline Assessment

All participants provided informed consent before participating in the study. Participants completed a questionnaire about personal factors including age, gender, previous VR use, and competitiveness. Competitiveness was measured with a validated 12-item inventory scored on a Likert scale of 1–7 called the Multi-dimensional Competitive Orientation Inventory (MCOI) [18]. The MCOI has been validated in healthy adults in Hungary (aged 18–59 years) and is a way to measure competitiveness in terms of 4 different competitive profiles: lack of interest, anxiety-driven, hyper-competitive, and self-developmental [18]. A self-developmental competitive profile indicates that the participant perceives competitive scenarios to both assess and improve one’s performance. A hyper-competitive profile indicates that the participant is willing to win at all costs and is less concerned about personal growth and development in competitive scenarios as winning is the primary concern. The final 2 profiles, anxiety-driven and lack of interest, are associated with the avoidance of competitions. An anxiety-driven profile is associated with feelings of anxiety in competitive scenarios. A lack-of-interest profile is associated with the absence of enjoyment or sense of meaning in competitions [18]. Though the MCOI does not have any known cutoffs by which individuals can be categorized as having one of these competitive profiles, participants in this study were classified based on the mean scores for each of the 4 subscales. Participants were classified as having a certain competitive profile based on their highest of 4 domains of the MCOI. If there was a tie, participants were classified as having a mixed competitive profile [18].

#### 2.4.2. Familiarization

Participants donned the HMD and bicycled in the three different conditions until they executed each cycling task correctly, were accustomed to the HMD, and understood the Borg Scale for RPE (Rating of Perceived Exertion) assessment [32]. After familiarization, there was a brief period of rest to restore HR to baseline before starting the 1st condition. See Figure 3 for an overview of the study protocol.

#### 2.4.3. Cycling Conditions

Baseline cycling cadence was established for each condition. The participants were instructed to cycle at a comfortable pace that they could maintain for 30 min. This comfortable baseline pace was used to set the target cadence (rate at which the visual markers were presented or rate of the virtual competitive cyclist) for the participants during the given trial. The target cadence was set at a pace that was 25% higher than comfortable baseline pace [14]. For the visual feedback condition, using a polynomial estimation of the road path allowed for the precise placement of markers within the VE. This polynomial estimation of the cycling path was also utilized to spawn other cyclists along the road at specific locations in the 2 competitive conditions. The participants were familiarized with each condition prior to data collection. Explicit instructions about the goal of each trial were provided before the trial began. The 3 conditions were counter-balanced, with competitive conditions in one block and visual feedback in the other block, as follows: Sequence A—Feedback, Competition (other), Competition (self). Sequence B—Feedback, Competition (self), Competition (other). Sequence C—Competition (other), Competition (self), Feedback. Sequence D—Competition (self), Competition (other), Feedback.

Visual Feedback: The cyclist was instructed to cycle at a pace that would turn the road markers blue. If the cyclist pedaled at a speed that was below the target cadence, the markers would remain white. If the cyclist cycled at a pace that was above the target cadence, the markers would turn from white to blue, serving as visual feedback for the user to maintain the pace above the target. Summary feedback about cadence was provided at one-minute intervals. The instructions to the participants were as follows: “The goal of this activity is to bicycle fast enough to make the markers on the road turn from white to blue”. This simulation and protocol design was iterated from previous work in the lab, implementing virtual bicycling simulations for persons post-stroke [33,34] and for persons with Parkinson’s Disease [14,35].

Competition against Others: The virtual cyclist used in both competitive conditions was controlled by the computer program [36]. The participants were instructed to cycle fast enough to pass the virtual agent who was traveling at their target cadence. Once the cyclist passed the virtual agent, another agent appeared further ahead on the road (traveling at the target cadence). The goal of the activity was to pass as many virtual agents as possible within the 5 min condition. The instructions to the participants were as follows: “The goal of this activity is to pass as many cyclists as you can”.

Self-Competition: This condition was similar to the previous competition condition. However, the participants were instructed that the virtual agent represented their best time. Therefore, the condition was framed as competition against oneself as the goal was to perform better than their best time by passing as many other virtual agents as possible. The instructions to the participants were as follows: “The goal of this activity is to beat your own time by passing as many cyclists as you can”. If this condition was the first of the 3 conditions (Sequence D), the participants were told that the other riders represented and were modeled after themselves using their cycling data collected during familiarization.

### 2.5. Outcomes

#### 2.5.1. Neuromuscular and Cardiovascular Intensity

Cycling cadence (RPM) data were collected at 1 Hz and averaged across the time series for each participant. Heart rate data collected at 1 Hz were averaged and normalized for age for each participant using the Karvonen formula for maximum HR (Max HR = 220—age) [37]. Based on each participant’s maximum HR, a percentage of maximum heart rate (%MaxHR) was calculated as (%MaxHR = average HR/max HR) for each condition.

#### 2.5.2. User Experience of Exercise

User experience of exercise, defined as motivation and enjoyment, was measured using the Intrinsic Motivation Inventory (IMI) total score and subscales [38]. IMI is a method to assess intrinsic motivation based on the SDT [7]. There are several subscales that are all related to different components of SDT. The primary subscale is the interest and enjoyment subscale as it measures feelings that are inherent to the task. It is closely related to autonomy and self-determination, as individuals are more likely to be intrinsically motivated to perform a behavior if they find it inherently interesting or enjoyable. Competence is another subscale that is positively correlated with intrinsic motivation and measures one’s self-assessment of the quality of their performance in a task. The value and usefulness subscale is important to consider when assessing the internalization of a given behavior, as individuals are more likely to be intrinsically motivated to perform a task in the long term if they understand and have internalized its value. Finally the effort and importance subscale is often considered a consequence of motivated behavior rather than a direct measure of intrinsic motivation, since individuals who are motivated to perform a behavior are more likely to expend more energy and effort in the behavior and perceive that it was important for them to work hard and perform their best on the task.

Particular subscales of IMI were chosen in congruence with the study purpose. The full subscale of interest/enjoyment was also collected (7 items) as this is a direct measure of intrinsically motivated behavior. Further, the full subscale of effort/importance was collected (5 items) and 2 items were collected each from the value and usefulness and perceived competence subscales. A total IMI score was calculated by averaging all 16 items, ranging from 1 to 7, with higher scores indicating higher motivation (see Appendix A).

RPE was measured with the Borg Scale which has been validated in the healthy population [32]. RPE scores range from 6 to 20 and higher RPE scores represent higher perceptions of effort. At the end of the session, participants ranked in which condition they worked the hardest and which condition they liked the most and offered comments.

#### 2.5.3. Visual Attention

Visual attention was operationally defined using several measures of task focus and visual exploration which were derived from eye-tracking data collected from wearable sensors integrated into the VR display. Eye-tracking data were collected through the HTC Vive Pro Eye (New Taipei City, Taiwan) at 50–60 Hz using the Vive SRanipal SDK (Version# 1.3.2.0) (Software Development Kit). Invalid eye-tracking data were removed from each dataset, by identifying datapoints where one or both of the eyes were closed (pupils not detected, e.g., during blinking). Valid gaze data were binned into different regions over the course of the entire trial, according to the following procedure. Dwell time percentage (DTP) on regions of interest was the primary method by which visual outcomes were generated from eye-tracking data [39]. The DTP quantifies what percentage of the total valid gazes was directed at different regions. It was derived from raw eye-tracking data, representing positions in space where participants were looking at any given time, collected at roughly 18 milliseconds (50–60 Hz). These gazes were mapped in space over the duration of the trial and visualized as a heatmap representing a spatiotemporal measure—a proxy for attention (see Figure 4). The points on this were binned to measure attention in two different ways, task focus and visual exploration [39].

Task focus was operationally defined as the % of total valid trial time that the participants spent looking at the road. The road region directly in front of the participant in the direction of travel was chosen as the location representing task focus as the stimuli relevant to the task; i.e., road markers (for the visual feedback condition) and the virtual cyclist (for the competition condition) were always located in this region. Visual exploration was measured through 3 separate variables. Rightward gazes are operationally defined as gazes directed to the right of the road (10 degrees off road center), roadside gazes are directed away from the road center either to the left or the right (30 degrees off road center, where the road is no longer visible in peripheral vision), and water gazes are directed toward a body of water in the simulation on the right of the track. All values are expressed as percentages of all valid gazes during each 5 min trial to normalize the values and control for the total number of valid gazes.

### 2.6. Data Analysis

Differences between conditions were first analyzed using a 3 × 1 one-way repeated-measures ANOVA (rmANOVA) for each outcome. The Mauchly Test was performed prior to the rmANOVA to assess sphericity, and a Greenhouse–Geisser correction was performed if the result of the Mauchly Test resulted in a *p*-value < 0.05. The rmANOVA tests were also adjusted using a Bonferroni–Holm correction for each category of variable (3 sub-measures of exercise intensity, 5 sub-measures of the user experience of exercise, and 4 sub-measures of visual attention) to correct for running multiple omnibus tests. Prior to performing post hoc tests, data were inspected for normality. Normal distribution was determined with Shapiro–Wilk Tests (*p* < 0.05) and visual inspection. Nonparametric post hoc Wilcoxon Ranked-Sum Tests were performed instead of paired *t*-tests when the data failed normality tests. When performing post hoc tests, a Bonferroni-corrected value of alpha = 0.0167 (0.05/3) was utilized to adjust for 3 separate post hoc analyses. All statistical analyses were performed using the Statistical Package for Social Sciences (SPSS, Version# v29, Chicago, IL, USA).

## 3. Results

### 3.1. Demographics

Participants included 25 healthy adults (18 male) with a mean age of 26.5 years old (22–34). Sixteen had fully immersive virtual reality experience, and nine bicycled regularly for exercise. Following the IPAQ (International Physical Activity Questionnaire) guidelines, 12 participants had high physical activity levels, and the remaining 13 participants had moderate physical activity levels [40]. The majority of participants had a self-developmental competitive profile (*n* = 19), followed by anxiety-driven (*n* = 3), mixed profiles (*n* = 2), and lack of interest (*n* = 1). For a full summary of participant characteristics, see Table 1.

### 3.2. Exercise Intensity

Age-adjusted heart rate was significantly different across the three conditions. Competition conditions had higher heart rates than feedback conditions (both *p* < 0.01). There were no statistically significant differences between the two competitive conditions (W = 118; *p* > 0.0167). Cycling cadence was significantly different across the three conditions. Faster cadences were observed in the competitive conditions compared to feedback (both *p* < 0.01). There was no difference between the competitive conditions (t(24) = −1.92; *p* > 0.0167). See Figure 5 for plots of raw cadence and age-adjusted HR for each condition. See Appendix A for data from the statistical analyses.

### 3.3. User Experience of Exercise

The total score of IMI was significantly higher in the two competitive conditions compared to feedback (both *p* < 0.01). There were no statistically significant differences between the two competitive conditions (*p* > 0.0167). The effort subscale of IMI was higher in both competitive conditions compared to feedback (both *p* < 0.01). There were no statistically significant differences between the two competitive conditions (*p* > 0.0167). The enjoyment subscale of IMI was significantly higher during self-competition compared to feedback (W = 32.5; *p* = 0.007). There were no statistically significant differences between feedback and competition against others or between the two competitive conditions (both *p* > 0.0167). The change in RPE from start to end of trial was significantly different across the three conditions. RPE was significantly higher in the competitive conditions compared to feedback (both *p* < 0.001). There were no statistically significant differences between the two competitive conditions (*p* > 0.0167). In post-session debriefing, 16 of the participants (64%) said they enjoyed self-competition the most. Additionally, 21 of the participants (84%) said they worked the hardest in self-competition. See Table 2 for a summary of measures of the user experience of exercise in each condition. See Appendix A for data from the statistical analyses.

### 3.4. Visual Attention

Across all participants and all three conditions, 92.87% of the gaze datapoints were valid, representing about 21 s of invalid data per 5 min trial, which includes invalid gazes secondary to blinking. Using the Bonferroni–Holm correction to adjust for running four repeated-measures ANOVAs, there were no statistically significant differences in any of the measures of visual attention across the three conditions. Therefore, individual post hoc paired *t*-tests are not included in the results. See Figure 6 for plots of visual attention in each condition. See Table 3 for a summary of measures of visual attention in each condition. See Appendix A for data from the statistical analyses.

## 4. Discussion

### 4.1. Exercise Intensity

Consistent with our hypothesis, cardiovascular and neuromuscular exercise intensity were both higher in the competitive conditions compared to visual feedback. On average across all young adults, cardiovascular exercise intensity was at least 68% of age-adjusted maximum HR and average cadence exceeded 100 rpm across all conditions. Only during the self-competition condition, did participants reach a vigorous cardiovascular intensity on average (77% or higher, as defined by the American College of Sports Medicine) [1]. Participants bicycled at an average cadence of 100–120 rpm across all three conditions. It is important to note that the purpose of this study was to drive high-cadence bicycling at lower resistances. Compared to high-resistance bicycling, lower resistances may reduce muscle fatigue so that participants can sustain aerobic exercise in their target HR zone for longer periods of time [41,42,43]. Therefore, this VR simulation may be valuable at promoting sustained aerobic exercise at a high cadence in a target HR zone, though longer bicycling bouts over longer time spans may be necessary to study effects on improving cardiovascular fitness.

### 4.2. User Experience of Exercise

Partially consistent with our hypothesis, intrinsic motivation and enjoyment were higher in the self-competition compared to visual feedback and competition against others conditions. However, contrary to our hypothesis, RPE was greater during competition. Across all three conditions, young adults reported high intrinsic motivation, considering that the total IMI score and several domains of IMI all exceeded the “somewhat true” anchor (score of 4/7) on IMI. Therefore, across all three conditions, young adults had high enjoyment, considered it important to expend energy in each task (effort and importance), and had high total motivation during all three tasks. Comparing the three conditions, young adults had a higher effort/importance rating in the two competitive conditions compared to the visual feedback condition, which is congruent with the findings that they worked harder and reported a higher perceived exertion in these two competitive conditions. Further, the total IMI score and the IMI interest/enjoyment score were both higher in self-competition compared to visual feedback. However, there was no difference in enjoyment comparing visual feedback to competition against others. These findings may be partially explained by the competitive profiles of the young adults, as the majority of participants had a self-developmental competitive profile, which could be related to their higher enjoyment to exercise only in the competitive condition that was framed as competing against oneself.

Although there were no differences in exercise intensity or perceived effort between the two competitive trials, participants ranked self-competition as the most strenuous of the three conditions (88%, *n* = 22). Similarly, though there were no differences in any of the user experience outcomes between the two competitive conditions, the majority of participants (64%, *n* = 16) liked the self-competition condition the most. These findings indicate that participants preferred the self-competitive condition when reflecting on the session after experiencing all three conditions, despite no differences in motivation and enjoyment between the two competitive conditions using IMI.

There are important implications of these findings that may be related to the SDT and the subscales of IMI used in the study. Higher effort and importance scores might be related to higher motivation to bicycle in the competitive conditions. Higher scores on the interest and enjoyment subscale of IMI (a direct measure of intrinsic motivation) suggest that participants had high autonomy and feelings of self-directed behavior in the competitive conditions. Those who were motivated to complete the task may have found that the successful completion of the task was important to them. This higher importance in completing the task may have resulted in faster cycling cadences, as pedaling faster would result in passing more riders on the road. The increases in HR (higher cardiovascular intensity) are directly related to the higher neuromuscular intensity and may have also resulted in higher perception of effort. Only in the self-competitive condition did young adults work at a vigorous exercise intensity, and this was the only competitive condition that had significantly higher scores on the interest/enjoyment subscale of IMI. Therefore, higher interest/enjoyment could reflect an increase in intrinsic motivation to accomplish the task associated with faster pedaling and cardiovascular demand.

It is important to consider that most individuals in the study were moderately to highly active according to their scores on the IPAQ and had relatively high baseline cadences (averaging over 68 RPM at their comfortable cycling cadences). Furthermore, in the visual feedback condition, despite pedaling at over 100 rpm on average and having a high average HR (68% of Max HR), participants still reported a 13 on the RPE scale, a measure corresponding to a “light” perception of effort. This low reporting of the RPE might be partially explained by the physical activity profiles of participants, as individuals who are accustomed to routine intense exercise may not perceive exercise at this intensity to be difficult or challenging. However, in the competitive conditions, participants averaged an RPE score corresponding to a “hard” perception of effort, reflecting the higher cadences (115–120 rpm) and HR (75–78% of maximum HR). Therefore, measures of exercise intensity, enjoyment, and motivation may be related to the fact that the study population consisted mostly of physically active young adults.

Though larger samples and more analyses would be required to study the relationship of competitive profiles to motivation to exercise, the trends from the findings in this study regarding the user experience of exercise and competitive profiles may be congruent with findings by Song et al., 2013, where individuals who were classified as “high competitive” preferred the competitive over the non-competitive exercise condition [44]. This study used a custom four-item questionnaire with items related to how participants felt about competition in general. The students were classified as either “high competitive” or “low competitive” and then randomly assigned into two groups, one where they competed against others in exergames and another where they played exergames alone. Participants classified as highly competitive enjoyed the competitive condition more, were intrinsically motivated to continue playing the exergames, and rated their gaming experiences more favorably compared to their non-competitive counterparts [44]. Findings from this study are also consistent with those from an article by Michael et al., 2020, in which young adults bicycled in a VR condition that is most similar to the self-competition condition implemented in this study [17].

### 4.3. Visual Attention

Partially consistent with our hypothesis, task focus was higher in the competitive conditions compared to the visual feedback condition, though these findings were not statistically significant. There were also no significant differences in measures of visual exploration among the three conditions, though a trend is present showing lower visual exploration and higher task focus in the competitive conditions compared to visual feedback.

Visual focus on the task measured using eye-tracking data was high across all three conditions, with participants on average spending at least 73% of the trial focusing on the task. This finding suggests that the use of visual stimuli administered through fully immersive VR (namely, visual feedback markers and competitive riders) can focus attention during high-intensity exercise. Though task focus was higher in the two competitive conditions, there were no statistically significant differences across all three conditions, which may be partially due to the large variability in visual behaviors across participants.

Visual exploration was highest in the feedback condition, measured with roadside gazes and gazes specifically on a body of water in the scene, which was accompanied by chirping birds and a parked bicycle by the waterside. This finding of lower visual exploration during the conditions in which participants were exercising at a higher intensity could reflect a tendency of participants who exercise at a higher intensity (or perceive they are working harder) to explore the environment less, and vice versa.

Overall, findings from this study suggest that visual attention may not be the primary underlying mechanism driving exercise intensity, enjoyment, and motivation. However, one interesting observation from the data is the trend in higher task focus in the conditions where participants exercised at a higher intensity, namely, the two competitive conditions. Participants also tended to favor these two competitive conditions in which they exercised at a higher intensity and focused more on the task at hand, possibly suggesting an interplay of exercise intensity, the user experience of exercise, and task focus. The relationship among these outcome measures may further be explained by participants’ competitive profiles, as most participants had a self-developmental competitive profile, which could promote working harder in the competitive conditions and focusing more on the task.

### 4.4. Limitations and Future Directions

One of the main limitations of the study is the measurement of motivation over a short timescale (three 5 min bicycling bouts). High measures of intrinsic motivation were reported across all three conditions, but these findings could be related to the novelty of the VR simulation and the bicycling tasks. Understanding intrinsic motivation during longer bouts of competitive VR bicycling can also provide insight into adherence to exercise, as sustained exercise (multiple sessions per week) is important to achieving and maintaining the health benefits of exercise [1].

Future directions include exploring the relationships between competitive profiles and study outcomes to provide further insight into the relationship between competitiveness and motivation to exercise. The high prevalence of self-developmental competitive profiles among participants in this study may be related to the higher exercise intensity in the self-competition condition, their motivation to exercise, and their history of engaging in moderate and vigorous intensity regularly. Studying other populations, such as older adults and participants with different health conditions, will extend our understand of VR as a tool to increase exercise intensity, motivation and visual attention with aging and disease.

## 5. Conclusions

Bicycling in fully immersive virtual environments promoted moderate cardiovascular and neuromuscular exercise intensity in both visual feedback and competitive conditions. Participants preferred the competitive conditions and only reached a vigorous exercise intensity during self-competition, which was congruent with their self-developmental competitive profiles. Measures of visual attention validated a task-relevant focus of attention. While there were no significant differences in task focus or any measures of visual attention across all three conditions, a trend is present showing higher task focus and lower visual exploration in the competitive conditions. This study also demonstrated that the use of a custom VR system, implementing various wearable sensors to monitor visual attention, exercise intensity, and perception of effort in real time, is tolerable during short bouts of bicycling in VR.

## Figures and Tables

**Figure 1 sensors-24-06873-f001:**
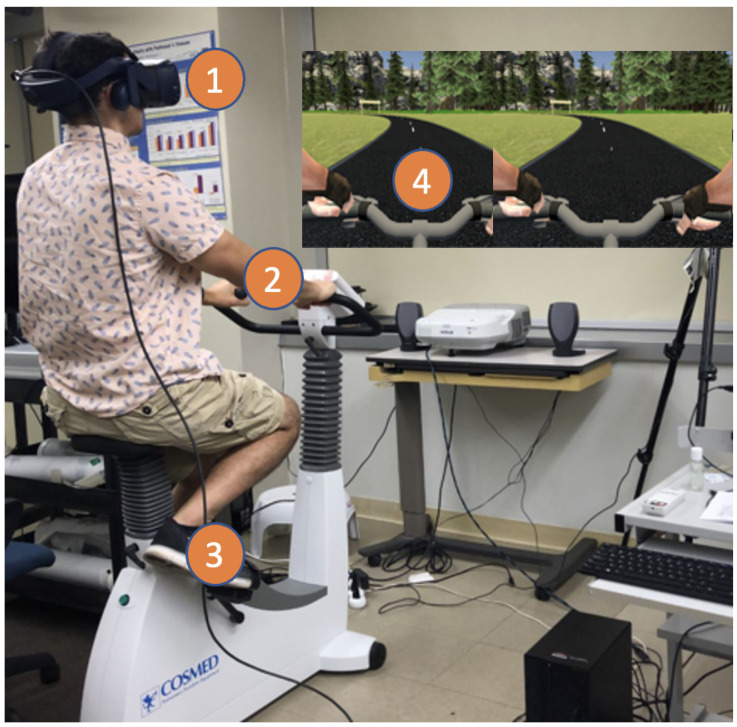
The experimental setup is shown with markers to denote how outcome measures were collected in the protocol. (**1**) Eye-tracking collected as a surrogate of visual attention using sensors embedded within the HTC Vive Pro Eye head-mounted display. (**2**) Heart rate measured using the Polar optical sensor. (**3**) Cadence measured with the Wahoo RPM sensor attached to the pedal crankshaft. (**4**) Perception of effort measured using a virtual Rating of Perceived Exertion (RPE) scale administered in the simulation and visible when the participant looked downward at the bicycle handlebars. The RPE scale could be toggled on and off by the protocol administrator.

**Figure 2 sensors-24-06873-f002:**
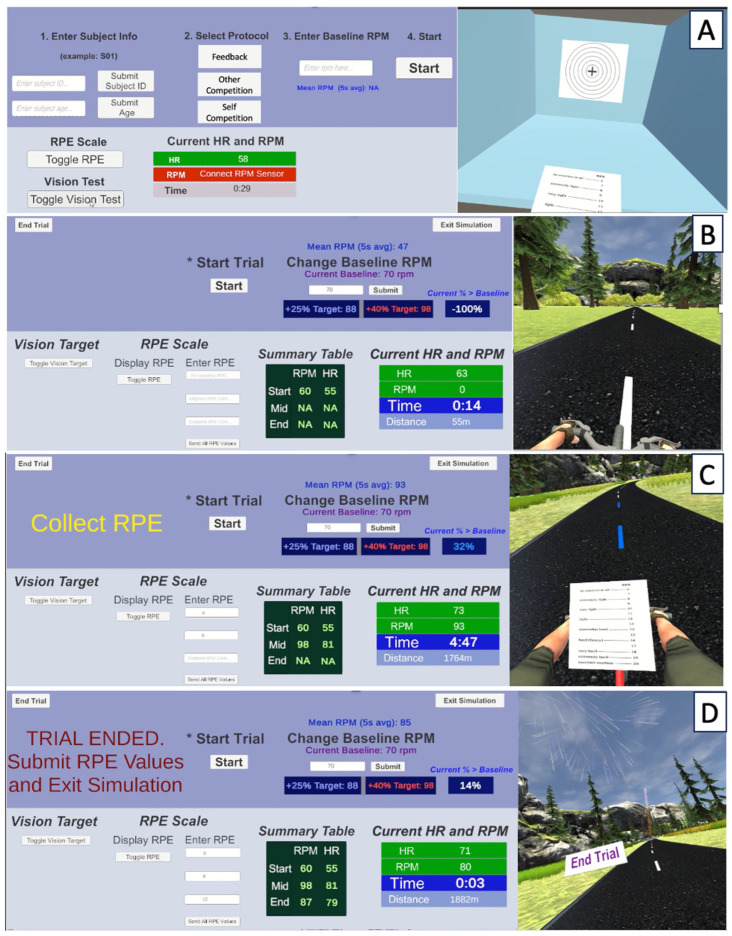
A flow diagram of the simulation, with the control panel shown on the left and the participant’s view of the simulation within the head-mounted display shown on the right. Panel (**A**): The pre-trial start screen where participant information is added and the protocol is selected. A vision test is shown to optimize the virtual reality headset visual settings for each user. Heart rate (HR) and cadence (revolutions per minute—RPM) are shown. Pressing the “Start” button (denoted with an asterisk to make the user interface more user-friendly) transitions the scene to the view shown in Panel (**B**). Panel (**B**): The participant now sees the virtual reality bicycling environment. The baseline cadence is manually entered into the text field and updates when the “Submit” button is pressed. After the baseline cadence is collected, the “Start” button is pressed and the simulation begins moving. Panel (**C**): At the midpoint and endpoint of the trial, the participant is instructed to look down at the handlebars and report their rating of perceived exertion score. Panel (**D**): When the participant finishes the trial, a box appears, letting them know the trial is complete. Fireworks and a finish line appear in the distance, and when the participant pedals past the finish line, the simulation ends.

**Figure 3 sensors-24-06873-f003:**
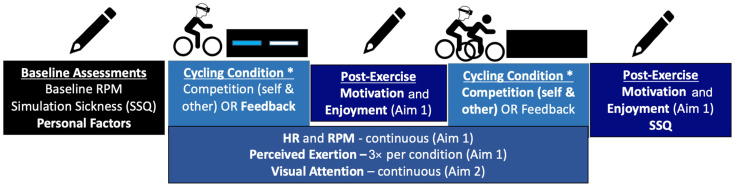
An overview of the study protocol in a single session. There are three trials: competition (self or others) and feedback. Single cohort, repeated measures. All participants complete all 3 conditions, which includes the 2 competitive trials (self-competition and competition against others) and visual feedback. The asterisk denotes that the 3 conditions were counter-balanced, described further in the manuscript text.

**Figure 4 sensors-24-06873-f004:**
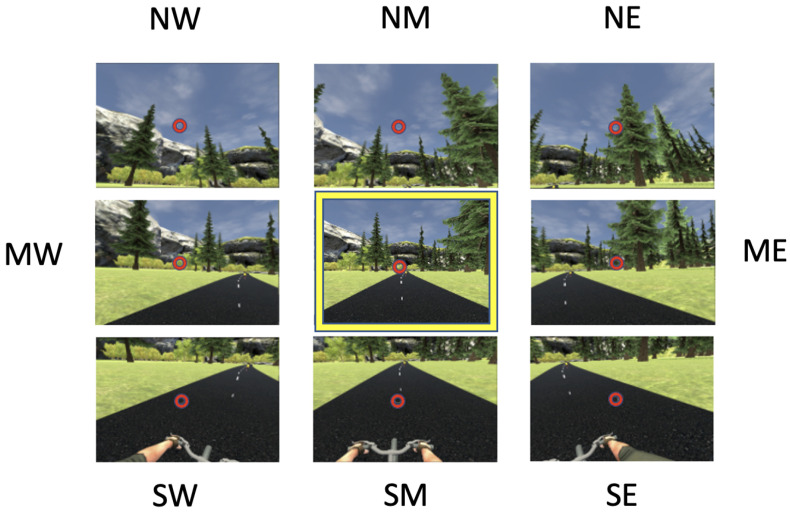
Breakdown of the virtual environment to categorize and analyze eye movements in VR. NW is the northwest region, NM is the north middle region, NE is the northeast region, MW is the middle west region, ME is the middle east region, SW is the southwest region, SM is the south middle region, and SE is the southeast region. The yellow square represents gazes focused on the middle of the road. The red circle in each box represents the location of the measured gaze.

**Figure 5 sensors-24-06873-f005:**
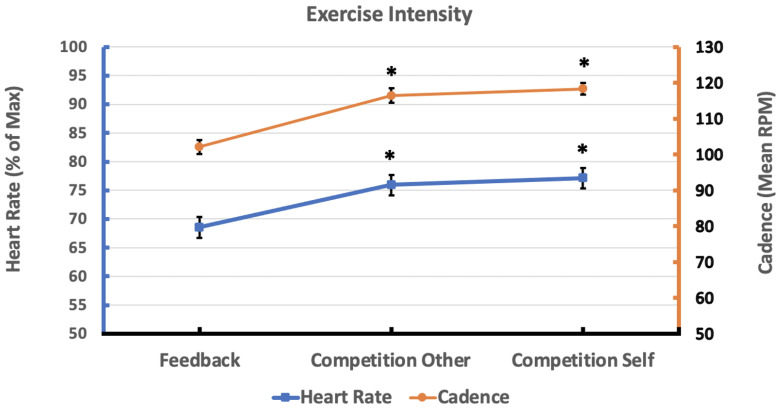
Exercise intensity in three conditions. The asterisk indicates statistically significant differences in each competitive condition compared to visual feedback (*p*-values < 0.0167). Cardiovascular intensity was higher in competition against others compared to visual feedback (t(24) = 4.74; *p* < 0.001) and in self-competition compared to visual feedback (W = 284; *p* < 0.001) Neuromuscular intensity was higher in competition against others compared to visual feedback (t(24) = 6.04; *p* < 0.001) and in self-competition compared to visual feedback (t(24) = 7.53; *p* < 0.001).

**Figure 6 sensors-24-06873-f006:**

Visual attention in three conditions. Task focus is operationally defined as the percentage of valid gazes falling in the center square. FOV stands for the field of view.

**Table 1 sensors-24-06873-t001:** Participant characteristics. Baseline cadence is a bicycling cadence (revolutions per minute—rpm) at which participants felt they would comfortably cycle for 30 min.

Characteristic	Value
N (female, male)	25 (7, 18)
Age (mean years, range)	26.5 (22–34)
VR Experience (*n*)	16(64%)
Bicycling Regularly	9 (36%)
Physical Activity (MET-minutes, std)	2943 (1810)
Low Activity (*n*)	0
Moderate Activity (*n*)	13
High Activity (*n*)	12
Baseline Cadence (rpm, std)	68.6 (4.0)

**Table 2 sensors-24-06873-t002:** Measures of the user experience of exercise for all participants across three conditions. Standard deviations are indicated in parentheses, and interquartile ranges are indicated for outcomes failing normality assumptions.

	Feedback	Competition (Other)	Competition (Self)
Endpoint RPE (/20)	13.0 (12–13)	15.7 (2) *	16.0 (15–18) *
Change in RPE (End–Start)	5.0 (1.8)	8.12 (2.1) *	8.80 (2.2) *
IMI Effort (/7)	4.8 (1.5)	6.0 (5.1–6.6) *	6.2 (5.7–6.8) *
Worked Hardest (Rank During Debriefing)	1 (4%)	3 (12%)	21 (84%) *
IMI Enjoyment (/7)	5.5 (1.0)	5.7 (0.9)	6.1 (5.0–6.8) *
IMI Total (/7)	5.4 (0.9)	5.8 (0.7) *	5.9 (0.8) *
Liked Most (Rank During Debriefing)	5 (20%)	4 (16%)	16 (64%) *

* Indicates significant differences compared to visual feedback condition (alpha = 0.0167).

**Table 3 sensors-24-06873-t003:** Measures of visual attention for all participants across three conditions. Task focus is a measure of how much time the participant spent focusing on task-relevant stimuli located near the center of the road. Visual exploration is measured through three separate variables. Rightward gazes are gazes directed to the right of the road, roadside gazes are directed away from the road center, and water gazes are directed toward a body of water in the simulation on the right of the track. All values are expressed as percentages of all valid gazes during each 5 min trial. Standard deviations are indicated in parentheses, and interquartile ranges are indicated for outcomes failing normality assumptions.

	Feedback	Competition (Other)	Competition (Self)
Task Focus (%)	73.4 (15)	77.7 (13)	79.1 (14)
Roadside Gazes (%)	1.90 (0.43–3.01)	0.59 (0.20–2.03)	0.59 (0.15–0.87)
Rightward Gazes (%)	7.4 (4.4–12.1)	6.5 (3.6–9.1)	4.3 (3.7–5.9)
Water Gazes (%)	0.153 (0–0.85)	0 (0–0.17)	0 (0–0.11)

Post hoc tests were not performed on any outcomes measures of visual attention as omnibus tests were not statistically significant.

## Data Availability

The original contributions presented in the study are included in the article/Appendix A, and further inquiries can be directed to the corresponding author. This article is a revised and expanded version of a paper entitled “Visual Attention in Virtual Reality: Comparing Visual Feedback and Competition in A Virtual Cycling Environment (VCYCLE-Competition)”, which was presented at the International Conference on Virtual Rehabilitation (ICVR 2022), Rotterdam, The Netherlands, July 2022 [45].

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
