# Peer review of "The Effects of Competition on Exercise Intensity and the User Experience of Exercise during Virtual Reality Bicycling for Young Adultsâ€"

_sensors, 2024, doi:10.3390/s24216873_

Round 1
Reviewer 1 Report
Comments and Suggestions for Authors
This paper shows the use of a custom bicycling VR system with wearable sensors to monitor visual attention, exercise intensity, and perception of effort in real-time. The paper is well written and has some interest although this is not a breakthrough. But, this is typically a paper that needs a video. I am ready to review again this paper if I can see a video demo.
Reviewer 2 Report
Comments and Suggestions for Authors
The article shows an interesting use case of VR application to enhance the engagement of participants in bicycling. The main focus of studies is aligned with increasing internal motivation and self-determination. The crucial factor was the competitivity of the user profile and how it is influenced by the virtual competitors showing off inside the application.
The experiment was arranged correctly and scientifically sound. The VR app simulated the motion of the bicyclist and was implemented in Unity for experimental purposes. I think that describing more details about app design and implementation, such as the used SDK, algorithms description, and diagram of the application flow, would significantly improve the impact of the article.
I would also suggest tha the authors include inside the article a photo of the experimental set-up, showing participants, bikes, VR headsets, etc.
In summary, I believe that VR technology has the potential to improve behavioural studies, ex[panding a wide range of fields for social experiments.
Reviewer 3 Report
Comments and Suggestions for Authors
The purpose of the current paper is to evaluate the effectiveness of fully immersive VR in promoting exercise intensity and adherence through visual feedback and competition among young adults. The results indicated that young adults bicycling outperformed the in VR condition demonstrating higher exercise intensity and motivation.
Indeed, the use of digital technologies especially immersive VR in exercise is a topic of increased interest. In addition, the paper follows are transparent methodology.
-The title includes the main concepts. However, I believe that it is lengthy. However, it is comprehensive and acceptable.
-The abstract is well-structured. The researcher clearly mentions the objectives the methods, the results, and the main conclusions.
-In the introduction section, the authors clearly summarize the current state of the topic. The limitations of current knowledge in this field are mentioned. The importance of the study is explained. The aims as well as the research questions and hypotheses are mentioned.
The study design and methods fit with the research question. The experimental procedure is adequately analyzed. The inclusion/exclusion criteria for the samples are described. The researcher followed a protocol with a wide range of tools to assure validity.
The results section presents results with accuracy providing relevant data.
Figures and tables are consistent with the data and the description within the text.
In the discussion section, the authors logically explain the findings comparing results with relevant studies. Limitations and future directions are mentioned.
In the text, reference numbers should be placed in square brackets [ ].
Acronyms/Abbreviations/Initialisms should be defined the first time they appear in each section.
Please check the capital letters in line 119.
Reviewer 4 Report
Comments and Suggestions for Authors
The present study investigated whether competition and/or visual feedback during VR bicycling impacted exercise intensity, user experience, and visual attention in a group of young, healthy adults. The authors found that both self-competition and other-competition increased age-adjusted heart rate and cycling cadence, compared to visual feedback. Both competitive conditions were linked with greater perceived effort compared to visual feedback, while self-competition led to greater enjoyment compared to visual feedback. Visual attention did not differ between the conditions. Overall, I found the present manuscript engaging, creative, and well-written. This study provides valuable insights into how VR technology can be used to promote exercise and would be of interest to many in the field. My only main concern is the lack of corrections for the number of rm-ANOVAs performed (please see more detailed comments and suggestions below). After addressing this, the manuscript only requires some minor edits to merit publication.
Introduction
- The introduction does not mention the two competition conditions used in the experiment: competition against oneself vs. competition against others. It would be helpful to mention this in the overview of the study design at the end of the introduction to help orient the reader before moving to the methods section. A relevant hypothesis would also help motivate the inclusion of both competitive conditions in the experiment.
Methods
- Lines 108-113: Well-done performing an a priori power analysis based on the results of previous literature.
- Line 113: Please include where participants were recruited from.
- Line 184: Why were only “14 participants familiarized with each condition prior to data collection”?
- Figure 1: If each participant participated in all three conditions, as indicated by the counterbalancing orders in lines 187-190, why are only two cycling conditions pictured here? This depiction makes it seem like participants carried out either the self or other competition condition, but not both. Please clarify.
- Line 208: If participants were instructed that the virtual agent represented their best time in the self-competition condition, what were participants told if this condition came first in the counter-balancing order?
- Lines 250-251: What percentage of the eye tracking data was considered invalid and removed from the dataset?
- Figure 2: Please include full descriptions of the abbreviations. For example, I’m pretty sure NM means north middle, but it would help the reader if this were explicit in the caption.
- Line 275: The authors state that they analyzed the results with “a 3x1 one-way … rmANOVA for each outcome”. There seem to be three main outcomes, each with their own set of sub-measures. Namely, for exercise intensity there were three sub-measures (i.e., heart rate, raw cadence, and normalized cadence), for user experience there were 5 sub-measures (i.e., change in RPE, endpoint RPE, IMI effort, IMI enjoyment, and IMI total), and for visual attention, there are 4 sub-measures (i.e., task focus, roadside gazes, rightward gazes, roadside gazes, and water gazes). This is a total of 12 separate rmANOVAs (as indicated by Tables S1, S3, and S5) without corrections for multiple comparisons. I appreciate that the post-hoc tests were Bonferroni-corrected, but the corrections for the number of main rmANOVAs should also be accounted for. For example, instead of running a 3 x 1 rmANOVA on each individual sub-measure, the authors could run a 3 (condition: self-competition, other competition, feedback) x [number of sub-measures] rmANOVA on each of the main outcomes separately (i.e., exercise intensity, user experience, and visual attention). For example, for exercise intensity, conduct a 3 (condition: self-competition, other competition, feedback) x 3 (sub-measure: heart rate, raw cadence, normalized cadence) rmANOVA. Another possibility would be to conduct a MANOVA with all three main outcomes included as dependent variables. Alternatively, the authors could apply a Bonferroni-Holm correction on their current results to address the issue of multiple comparisons. Most of the p-values and effect sizes, apart from some of the visual attention findings, look strong so I don’t expect this to change the results by much. However, adding a correction for the number of ANOVAs performed would strengthen the statistical rigor of the present manuscript.
- Line 275 continued: What statistical software package was used to analyze the results?
Results
- Line 316: Please include references to tables S3 and S4 in this section.
- Line 327: Please include references to tables S5 and S6 in this section.
Discussion
- Line 333-335: The authors state that “only during the self-competition condition did participants reach a vigorous cardiovascular intensity”. This is confusing since there were no statistically significant differences in age adjusted heart rate or cadence between the self-competition and other-competition conditions (lines 294-301; Figure 1). Please rephrase for clarity.
- Line 384:-395: The authors do a good job of describing a key limitation of the study.
Minor points:
- Figure 3 caption: “p < .0167” should be “p’s < .0167” since there are several asterisks.
- Line 319: “Roadside gazes were significantly differences …” should be “roadside gazes were significantly different …”.
- Line 412: The authors point out in the introduction that visual attention may be the underlying mechanism explaining exercise intensity (line 83). However, the present study did not find any differences between the conditions in terms of visual attention, despite observing that the competitive conditions boosted exercise intensity compared to feedback. It seems like the competitive profiles of the participants are the main driver of the increased exercise intensity in the competitive conditions, rather than visual attention. I wonder if the authors could explicitly connect these points to tie the manuscript together more tightly.
Round 2
Reviewer 1 Report
Comments and Suggestions for Authors
Thank you for the videos, this is nice. Now, I consider your paper as completed.